# Cocaine Effects without Cocaine: Increasing Happiness with Self-Regulation Therapy in a Single Session

Salvador Amigó 

Department of Personality, Evaluation and Psychological Treatments, Faculty of Psychology,
University of Valencia, 46010 Valencia, Spain; salvador.amigo@uv.es

**Abstract:** The psychological reproduction of cocaine's ability to increase happiness was studied. The first part of this study consisted of building and validating an instrument to measure the stable and transient aspects of happiness via the 10-item Happiness Trait–State Scale (10 HTSS) in a sample of volunteers ($N$ = 128). In the second part, Self-Regulation Therapy (SRT, a procedure based on suggestion and classic conditioning) was applied to all the participants to increase their relaxation during a single session. The relaxation session slightly increased happiness. A subgroup of participants who were cocaine users ($N$ = 33) took part in a second session to reproduce the stimulant and euphoric cocaine effects. That was a "mental" reproduction session to simulate the effects of cocaine. For both conditions, all the participants filled in the 10 HTSS at the beginning and the end of the session to compare scores and to prove if the happiness state increased. For the cocaine condition, happiness markedly increased, especially in comparison to the increase during the relaxation session, for both the total group and the cocaine users group. Although this increase was achieved during a single session, similar previous studies with more continuous training and monitoring suggest that improvement in emotions can be durable and long term.

**Keywords:** cocaine; happiness; depression; self-regulation therapy; euphoria; relaxation

## 1. Introduction

Cocaine is a well-known addictive and abused drug [1,2], but is accepted in medical use, especially as an effective safe anesthetic in surgical nasal, ear and throat procedures [3]. Recently, a study found that topical 4% and 8% cocaine is an effective anesthetic that can be safely administered for nasal procedures based on a randomized phase III clinical trial. This study aims to favor approval by the United States Food and Drug Administration (FDA) because many otolaryngologists frequently utilize topical cocaine in their practice [4]. For example, in a large survey of active members of the American Academy of Otolaryngology—Head and Neck Surgery, which was published in 2004, fifty percent of respondents reported using cocaine as a topical agent during endoscopic sinus surgery over the preceding calendar year [5]. In a survey of members of the British Association of Otolaryngology—Head and Neck Surgery, sixty-seven percent of all the surgeons regularly used cocaine [6]. This use of cocaine is currently placed in Schedule II in the Comprehensive Drug Abuse Prevention and Control Act of 1970 [7]. This law, commonly known as the Controlled Substance Act (CSA), establishes a federal policy in the USA to regulate the manufacturing, distributing, importing/exporting and use of regulated substances. Originally, the purpose of the CSA was to comply with the requirements of the Single Convention on Narcotic Drugs (1961), but the drug classification that the CSA proposes remains an international reference worldwide.

When determining the dependence potential, the classification of psychoactive drugs is normally based on their mood-altering properties or their subjective effects. Hence, cocaine has been proven to increase the acute effects of euphoria and induce a "high", with a course of action that depends on the dose and administration route [8]. A 25 mg intranasal

administration leads to significant changes in the high and pleasantness scales [9], and oral and intravenous cocaine produces positive subjective drug effects and increases the "good effects", "rush", "drug effect", and "liking" ratings [10,11]. Thus, both oral and intranasal cocaine produce comparable behavioral and subjective effects [12].

Even though cocaine produces subjective positive effects, it is not useful in psychology and psychiatry because of the potential abuse and adverse effects, and also because acute within-session tolerance develops during repeated intranasal cocaine administration [13]. Some side effects, risks and dangers of the medical and psychiatric consequences of use cocaine and abuse are addiction; depression; acute myocardial infarction and cerebrovascular accident; toxicity involving neurologic, obstetric, pulmonary, dermatologic, and gastrointestinal systems; and even sudden death [14,15]. In particular, the long-term effects of using cocaine can produce movement disorders, Parkinson's disease, and irritability and restlessness from cocaine binges, and some people also experience severe paranoia [16]. This is why therapeutic cocaine use in psychiatry is not currently accepted by the scientific community, and there is very little anecdotal evidence for its psychotherapeutic use. Anxiolytic-like effects after first exposure in animal models research [17,18] and a sense of well-being, calmness, and feelings of elation with intravenous infusions of 2.5–25 mg of cocaine in depressed patients have been reported [19]. Oral cocaine neither functions as a reinforcer nor does it significantly affect performance. Moreover, 50–300 mg is well tolerated by individuals with recent histories of cocaine use, and can be administered safely under controlled laboratory and medical conditions [10] (p. 11).

If cocaine can increase well-being and elation emotions, we can contemplate that it also has the potential to increase happiness. As previously pointed out, euphoria is one of the main subjective effects of a single dose of cocaine [8,20]. If we consider euphoria to be an intense transient emotion of happiness [21], then we can increase some people's happiness by eliciting this emotion at different times.

However, employing cocaine in a clinical setting to increase positive emotions might be difficult if we consider these dangerous and adverse effects. A psychological technique exists that allows it to be used because it is based on reproducing the same cocaine effects, but without adverse effects, in a safe environment. This technique is Self-Regulation Therapy (SRT) [22], which is explained later.

In clinical practice, and as a result of scientific research, it has been found that the patients who apply SRT can feel euphoria and well-being as often as they wish and it could, therefore, increase their happiness [23–27]. First, however, we need to measure happiness and have a sample of people to whom this technique has been applied for this purpose. All these questions are dealt with in this article.

There is scientific evidence for the possibility of treating drug addictions with psychological and "mental" procedures, such as yoga, meditation, cognitive behavioral therapy, etc. [28–31]. However, this is not a study on addiction treatment. It is fundamentally based on reproducing the effects of drugs by attempting to replace the drug itself and using these effects in an off-beat way to improve therapeutic abilities; that is, employing the positive effects of drugs, such as cocaine, to obtain a psychotherapeutic benefit.

This article is divided into two studies. The first one consists of creating and validating a post hoc Trait–State Happiness Scale from a list of mood adjectives used in this and previous studies. The entire sample (*N* = 128) participated in this first study. The sample contained people from the general population with different relationships with drugs, from abstinents, only cannabis users, and users of other drugs, both moderate and frequent. An Exploratory and Confirmatory Factor Analysis was used to obtain the scale factor structure in trait and state forms, as well as Pearson correlations to obtain convergent and discriminant validity.

In the second study, SRT was used to produce relaxation in the entire sample to test whether the relaxation condition increases happiness state scores. Subsequently, a subsample of 33 cocaine users was obtained. SRT was applied to this group to mentally reproduce the effects of cocaine, and scores on the state of happiness were recorded. Finally,

both happiness state scores were compared in this group to check which condition, whether relaxation or cocaine, produces a greater increase in the state of happiness. In this second study, different repeated measure ANOVA procedures were used to compare relaxation and cocaine conditions.

The methodology and analysis used in these two studies will be discussed in detail later on.

The main hypothesis of this study that it possible to first obtain a reliable measure of the trait and state of happiness and, second, to increase this emotion by reproducing cocaine effects during a stimulating session rather than a relaxing one.

Specifically, the hypotheses of this study are:

(1)   It will be possible to obtain a reliable measure of the trait and state happiness.
(2)   Applying SRT to produce relaxation in a single session will increase the state of happiness in a group of volunteers.
(3)   In a subgroup of this sample, made up of cocaine users, the relaxation session was followed by a session of reproducing cocaine effects with SRT. We expect the reproduction of cocaine effects to increase the state of happiness more than relaxation.

The experimental design to confirm, or not, these hypotheses will be presented in full detail later on.

## 2. First Study: The 10-Item Happiness Trait–State Scale (10 HTSS)

### 2.1. Introduction

The first thing is to answer the question about how happiness is measured, and even what happiness is. Measurement of happiness is usually based on the self-reports of subjective well-being, which is related to not only a high satisfaction with life, but also to the predominance of positive and pleasant emotions and moods and infrequent negative effects or unpleasant emotions [32]. Happiness has also been defined as "a lasting, complete and justified satisfaction with life as a whole" [33] (p. 16) and as "the preponderance of positive affect over negative affect with a distinct focus on the affective evaluation of one's life situation" [34] (p. 545). Three main components of happiness have been identified: frequent positive affect or joy; a high average level of satisfaction over a period; and not having negative feelings, such as depression and anxiety [35]. There are many concepts related to happiness, such as life satisfaction, flow, peak experiences, well-being, quality of life, and a wide variety of measuring instruments [36–38].

Debate exists about the stable or unstable nature of happiness, or happiness as a temperamental disposition (trait) or a variable emotion influenced by life circumstances (state). Veenhoven [39] concluded that happiness is quite stable in the short term, but not in the long run, and is not entirely built-in on a genetic basis, but is strongly influenced by environmental circumstances. No matter what the relation is between emotions, moods, and happiness, understanding the nature and measure of happiness is an essential and basic question. Different time scales or hierarchical level structures can be used to relate emotions and moods. Emotions can be considered as short-term experiences related to the affective events of everyday life at a bottom-up level, unlike happiness, which is thought to be regulated by rational thought and other high-level top-down cognitive processes. In a way, happiness can be conceived, at least in the Western mentality, as an accumulation of successive positive affective experiences, and a chance to sustainably increase happiness, which is possibly underpinned by emergent (bottom-up) sources [40].

Hence, the relation between mood states and happiness, at both different temporal and spatial levels, leads us to consider the trait–state happiness concept to be a legitimate alternative for understanding and measuring happiness. This is the purpose of this part of the article.

Different scales have been proposed to measure the stable and temporal dimensions of happiness. The Depression–Happiness Scale [41] is a 25-item self-report scale that contains 12 items concerned with positive thoughts, feelings, and bodily experiences, and 13 items with negative ones. According to these authors, this scale can be suitably employed with the

general population rather than a clinical one and may be useful for thinking of depression and happiness as opposite ends of a single continuum. The test–retest data over 2 weeks revealed considerable stability, which suggests that this scale might be taken as a trait measure of happiness rather than as a state measure [42], and even 2 years later [43].

Another alternative is the State/Trait Depression Inventory (STDS) [44]. This instrument measures two aspects of depression by evaluating the frequency (trait) and the degree of involvement and the intensity of a current emotion (state) to separately assess depression as a personality trait and as an emotional state in non-clinical samples [45]. This instrument includes two factors that represent the presence (dysthymia) and absence (euthymia) of depression. This latter component has reversed scores that represent a lack of positive affect. Each component is composed of five items and both components are correlated.

Other authors [46] consider that the current consensus about the bipolarity of the structure of affect, especially the bipolarity of happiness–sadness, as poles of an overarching bipolar dimension [47,48] is premature. They find that happiness and sadness are not bipolar opposites and that they require a two-dimensional model of affect to be fully understood.

The State–Trait Cheerfulness Inventory (STCI), with a Trait Version (STCI-T) [49] and a State Version (STCI-S) [50], measures three components (cheerfulness, seriousness, and bad mood) as the temperamental basis of humor. The Trait Version represents a stable personality characteristic, and the State Version denotes variable changes based on situational and contextual factors. The standard version comprises 30 items, with 10 items measuring each factor.

For the purpose of this article, a post hoc trait–state happiness scale was constructed based on a list of adjectives and an Exploratory Factor Analysis performed in previous studies. It also explored the relations between happiness, personality, and drug use.

A significant relation between the Eysenck Personality Questionnaire [51] and the Oxford Happiness Inventory [35] has been found, with a significant positive relation between happiness and extraversion and a negative one between happiness and neuroticism [52–54]. When the Oxford Happiness Inventory [55] and the NEO-FFI [56] are used, all the correlations between scores on the subscales of both measures were positive and significant ($p < 0.001$) for agreeableness, conscientiousness, extraversion, and openness to experience, and negative and significant for neuroticism.

Furthermore, in a sample of Australian adolescents, a positive correlation was found between satisfaction of life and extraversion and a negative one with neuroticism [57].

Gale et al. [58] examined the effects of neuroticism and extraversion of people aged 16 and 26 years old on mental well-being and life satisfaction about 40 years later. Extraversion had direct and positive effects on both the well-being measures, while neuroticism impacted both well-being and life satisfaction through susceptibility to psychological distress and physical health problems.

Cheng and Furnham [59] found that extraversion and an optimistic attributional style in positive situations were strong predictors of self-reported happiness.

Regarding personality and drugs, psychoticism and neuroticism have been found to be significantly higher in alcoholics and drug addicts compared to non-alcoholics and non-drug addicts [60].

Most participants score higher for psychoticism (100%) and introversion (75%), while the neuroticism (58%) trait appears less among drug abuse cases [61]. Psychoticism from Eysenck's dimensional model of personality is a key personality factor related to drug and alcohol users in non-clinical samples like student drug misusers [62,63]. In a large sample of 13–15-year-old British adolescents, Francis [62] found a significant relation between the Junior Eysenck Personality Questionnaire (JEPQ) and attitudes toward substance use. The more tolerant people to substance use scored high for psychoticism and extraversion and low on the neuroticism and Lie scales (socially non-conforming).

When comparing heroin addicts to the recreational drug users group, a higher extraversion and social conformity in the latter group were observed [64]. When comparing a

drug misuse group to occasional users and non-users [65], significant differences appeared for psychoticism, with higher scores obtained by the people with drug misuse than those who did no use drugs, and higher levels on the Lie scale for the people with drug misuse disorders compared to occasional users and non-users.

With a large sample of university students from first courses, regular drug users scored significantly higher for psychoticism and non-conformity (on the Lie scale) of the EPQ than moderate or non-drug users [66]. Rosenthal et al. [67] found that young cocaine users were more impulsive and less anxious than other kinds of drug users like opioid and alcohol users. Williams [68] reported that the combination of high neuroticism and low extraversion was associated with a more negative and less positive average mood and some greater mood variation, whereas low neuroticism and high extraversion led to a better stable mood, which proved the pervasive effects of extraversion and neuroticism on a positive negative mood. Finally, all four groups of drug users scored significantly higher for psychoticism than the comparison groups [60].

*2.2. Materials and Methods*

2.2.1. Participants

This study included 128 participants (52 males and 76 females) who were students (69.5%), employees (28.1%), and unemployed people (2.3%) from different cities of Spain. Their mean age was 24.75 (*SD* = 5.15) years and their age range was 19–50 years.

The participants were at first student volunteers from the University of Valencia (Spain). By the "snowball" procedure, they were informed about the existence of this study, which allowed other young students and workers from different cities of Spain to participate in the study.

Both studies 1 and 2 were carried out at the Faculty of Psychology of Valencia (Spain) in a training program for students to learn how to apply SRT. The application of SRT and the psychometric instrument administration were mainly in charge of the author of this article, who guided the application of SRT by the students in some cases.

The total sample had to be varied and include people from the general population with different relationships with drugs. The participants in this study were divided into four groups according to their drug use:

Group 1 (G1): never used illegal drugs (*n* = 27).

Group 2 (G2): only consume cannabis as an illegal drug (*n* = 41).

Group 3 (G3): moderate drug users (*n* = 37); drug use less than 30 times in life and less than 30 times in the last year (ecstasy, cocaine, and amphetamines).

Group 4 (G4): regular and heavy drug users, especially of stimulant drugs (*n* = 23), drug use more than 30 times in life and more than 30 times in the last year (ecstasy, cocaine, and amphetamines).

Both the first and second studies in this article were conducted in accordance with the Declaration of Helsinki, and the protocol was approved by the Ethics Committee of the University of Valencia (Spain) on 15 September 2017 (project identification code: H1499339130100). Additionally, informed consent was obtained from all subjects involved in the study.

2.2.2. Instruments

The Oxford Happiness Questionnaire-Short Scale (OHQ-SS) [53] is an 8-item Likert-type response scale. The scale score goes from 1 (strongly disagree) to 6 (strongly agree). This is the short-form version of the Oxford Happiness Questionnaire, a compact scale with 29 items for measuring psychological well-being.

The Substances Use Scale [69] follows the European Monitoring Centre for Drugs and Drug Addiction (EMCDDA) criteria. It is a brief self-report questionnaire that measures drug use frequency (i.e., cannabis, alcohol, tobacco, cocaine, MDMA, sedatives, hallucinogens, and amphetamines). People must answer questions like: Sometime in your life; How often in your life; How often in the last 12 months; How often in the last month.

The EPQ-RS [51] is a 48-item Yes/No response questionnaire that contains four sub-scales, and each one consists of 12 items, including extraversion, neuroticism, psychoticism and Lie.

The Depression Scale (DS) [68,70] is a 7-item Likert-type response scale with the following self-descriptive adjectives: blue, downhearted, sad, unhappy, troubled, useless, and worthless. The scale score goes from 0 (no effect) to 4 (maximum effect). Some changes in the selected adjectives were subsequently made based on the results of previous studies.

List of pleasant activation adjectives [71]. There are seven adjectives in the quadrant that correspond to pleasant activation from the multidimensional scaling of unipolar adjective scale ratings: peppy, active, energetic, enthusiastic, optimistic, in a good mood, and glad. Following the authors' instructions, the participants were asked to indicate the degree to which the adjectives described how they felt at that particular moment, but the scoring system was changed to unify criteria with the previous depression scale. Thus, the scale score went from 0 (no effect) to 4 (maximum effect).

### 2.3. Results

Data were analyzed using IBM Corp., released 2015, IBM SPSS Statistics for Windows, Version 28.0., IBM Corp. (Armonk, NY, USA). Exploratory and Confirmatory Factor Analyses were performed, and Pearson correlation coefficients were obtained for the convergent and discriminant validity of the 10 HTSS.

### 2.3.1. Exploratory Factor Analysis (EFA)

An Exploratory Factor Analysis (EFA) from the Depression Scale and List of pleasant activation adjectives items (14 in all) was performed with the main components and oblimin rotation (Delta = 0.8) to obtain two factors and to select item loading higher than 0.50 in the corresponding factor. The results presented in Table 1 show the Kaiser–Meyer–Olkin Measure of Sampling Adequacy, Bartlett's sphericity test, the correlation between components, and the percentage of variance accounted for by factors in both the trait and state versions. Three models for both the trait and state formats were obtained. The state format was obtained from the scores on the adjectives immediately before applying SRT during the first session. Cronbach's α scores for internal consistency are also shown.

**Table 1.** The Exploratory Factor Analysis results for the items from the Depression Scale and List of pleasant activation adjectives and Cronbach's α.

| Model | KMO | Bartlett | Rot Obli (0.8) N° Factors | Correlation between Components | % of Variance | α |
|---|---|---|---|---|---|---|
| Model 1 Trait | 0.839 | <0.001 | 2 | −0.74 | 63.03 | 0.84 |
| Model 2 Trait | 0.851 | <0.001 | 2 | −0.77 | 64.79 | 0.85 |
| Model 3 Trait | 0.845 | <0.001 | 2 | −0.76 | 62.93 | 0.84 |
| Model 1 State | 0.843 | <0.001 | 2 | −0.85 | 69.27 | 0.85 |
| Model 2 State | 0.836 | <0.991 | 2 | −0.85 | 66.85 | 0.84 |
| Model 3 State | 0.850 | <0.001 | 2 | −0.85 | 68.27 | 0.86 |

The adjectives for each model were Model 1: enthusiastic, peppy, optimistic, in a good mood, glad, blue, downhearted, sad, unhappy, worthless; Model 2: enthusiastic, peppy, optimistic, in a good mood, glad, blue, downhearted, sad, unhappy, troubled; and Model 3: enthusiastic, peppy, optimistic, in a good mood, glad, blue, downhearted, sad, unhappy, useless.

The high values in the Kaiser–Meyer–Olkin Measure of Sampling Adequacy and the low values in Bartlett's test of sphericity statistics for all the models indicate that a factor analysis could be useful with these data.

The factors obtained after rotation accounted for more than 60% of the variance for all the models. The correlation between factors was high and negative, with a close relation between them. We could interpret both factors as two happiness poles between pleasant

activation and depression. The Cronbach's α levels were more than acceptable for all the models.

### 2.3.2. Confirmatory Factor Analyses (CFA)

Then, Confirmatory Factor Analyses (CFAs) using Maximum Likelihood (ML) were performed to find the best structure of the test. All three possible models, and both the trait and state formats, were tested. They all included two factors, but adding Model 4 suggested two independent factors: pleasant activation and depression.

The following fit indices were obtained: Chi-square ($\chi^2$) in association with degrees of freedom (df), Comparative Fit Index (CFI), Root Mean Square Error of Approximation (RMSEA), Tucker–Lewis Index (TLI), and Akaike's Information Criterion (AIC).

The results of the CFAs concerning the loading values of latent variables on the observed variables are presented in Figure 1.

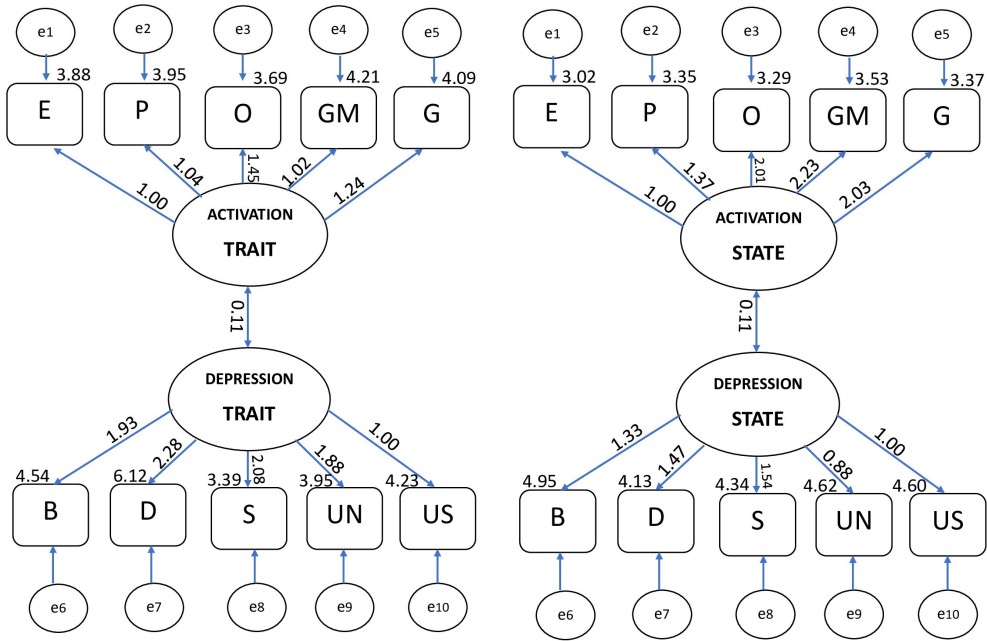

**Figure 1.** Confirmatory Factor Analysis of Model 3 for the 10-item Happiness Trait–State Scale (10 HTSS) in the trait and state formats. Note: E = enthusiastic; P = peppy; O = optimistic; GM = in a good mood; G = glad; B = blue; D = downhearted; S = sad; UN = unhappy; US = useless.

Table 2 shows the Goodness-of-Fit Indices for the assessed models. In general, significant $\chi^2$ values were obtained, which do not indicate a very good fit. However, $\chi^2$ was highly sensitive to sample size. Thus, we considered this index in association with others. In any case, low and non-significant $\chi^2$ values indicated an adequate fit present in Model 3.

**Table 2.** Goodness-of-Fit Indices in the assessed models.

| Model | $\chi^2$ | $p$ | df | CFI | TLI | RMSEA | AIC |
|---|---|---|---|---|---|---|---|
| Model 1 Trait | 72.41 | 0.000 | 34 | 0.93 | 0.89 | 0.054 | 133.41 |
| Model 2 Trait | 64.27 | 0.001 | 34 | 0.94 | 0.91 | 0.048 | 126.27 |
| Model 3 Trait | 58.31 | 0.006 | 34 | 0.95 | 0.93 | 0.043 | 120.31 |
| Model 4 Trait | 76.11 | 0.000 | 35 | 0.92 | 0.88 | 0.055 | 136.11 |
| Model 3 State | 43.73 | 0.112 | 34 | 0.98 | 0.97 | 0.027 | 105.73 |

The models in the trait format, especially CFI and RMSEA obtained for Model 3, indicated a good fit to the data compared to the other models. Model 2 showed an adequate fit for both RMSEA and CFI, but not a good a fit as Model 3. Likewise, the highest TLI values were for Model 3 (TLI = 0.93), which were especially good. Finally, Model 3 presented the

lowest values for AIC (120.31). Thus, only Model 3 in the state format was calculated. All the statistics showed a good fit, and were even better than Model 3 in the trait format.

In summary, Model 3 for both the trait and state formats showed the best fit indices, but we must take into account the high $\chi^2$ values.

2.3.3. Convergent and Discriminant Validity

The convergent and discriminant validity information was obtained by performing a correlation analysis between the 10 HTSS (both in trait and state format) and a subset of other scales. To show convergent validity, positive and significant relations between the 10 HTSS and the Oxford Happiness Questionnaire and extraversion should be expected. Discriminant validity was shown when low and non-significant relations between the 10 HTSS and neuroticism were obtained. The matrix correlation is presented in Table 3.

**Table 3.** The matrix correlation between the 10 HTSS and personality traits (EPQ-R).

|  | OHQ | 10HT | 10HS | E | N |
|---|---|---|---|---|---|
| 10HT | 0.65 *** | - | - | - | - |
| 10HS | 0.41 *** | 0.53 *** | - | - | - |
| E | 0.38 *** | 0.49 *** | 0.22 * | - | - |
| N | −0.53 *** | −0.64 *** | −0.42 *** | −0.30 *** | - |
| P | 0.00 | −0.07 | −0.02 | 0.03 | 0.01 |

Note: OHQ = Oxford Happiness Questionnaire; 10HT = 10-item Happiness Trait Scale; 10HS = 10-item Happiness State Scale; E = extraversion; N = neuroticism; P = psychoticism; * $p < 0.05$; *** $p < 0.001$.

The results are in line with what was expected. A positive statistically significant correlation between the happiness scores measured by the Oxford Happiness Questionnaire and 10-item Happiness Trait Scale format scales and all of them and extraversion were obtained. Furthermore, a negative statistically significant correlation between the happiness scales scores with neuroticism was found. The relation between extraversion and neuroticism was negative and statistically significant. ($r(31) = −0.30$, $p < 0.001$). No significant correlation between psychoticism appeared with any other variables.

We can conclude that the 10 HTSS showed convergent and discriminant validity in this study.

## 3. Second Study: Reproducing Cocaine Effects with Self-Regulation Therapy to Increase Happiness

*3.1. Introduction*

In this section, we checked to see whether it was possible to increase the state of happiness during a single session by reproducing the positive cocaine effects.

Self-Regulation Therapy (SRT) [22,72] is a suggestion procedure that is derived from a cognitive behavioral perspective of hypnosis [73], which applies techniques to reproduce physical sensations (weight in hand, salivation, etc.) with which patients can obtain a high degree of suggestibility to respond to any suggestion with their eyes open with a participative attitude, while maintaining a normal conversation with the therapist. This procedure is based on classic conditioning and therapeutic use of expectations and motivation. All this makes the technique very useful for reproducing drug effects.

SRT comprises three phases. In the first one, the subjects are asked to associate physical sensations (i.e., salivation, leg paralysis, arm heaviness, hand rigidity) with physical stimuli like images or words, which will help them to reproduce these sensations later with no physical stimuli. In the second phase, individuals have to reproduce these sensations several times until only verbal suggestion is necessary to produce physical sensations. The participants finally respond more easily and quickly while doing exercises. In the last one, the so-called generalization phase, any kind of demand has the suggested effects, such as different emotions and sensations that they have not practiced before, because their minds are completely receptive.

A detailed description of this procedure can be found in work by Amigó [22,74].

Classic conditioning of drug effects has been proven [75–79] and experimental drug conditioning designs in humans have been proposed [80].

Cocaine-conditioned drug effects have been proven in both rats [81–83] and humans [84–86]. In the first study with humans [84], cocaine-related cues caused reliable decreases in skin temperature and skin resistance and reliable increases in heart rate, self-reported cocaine cravings, and self-reported cocaine withdrawal in cocaine abuse patients.

Cocaine expectancy modulates subjective and objective responses to drugs [87,88].

SRT allows the effects of many different drugs to be reproduced [74]. There is scientific evidence for the potentiality of SRT to reproduce cocaine effects in a single case experimental design [89], a within-group design with a small group of cocaine addicts [27], and a within-group design with young stimulant drugs users [90,91]. In the first study [89], a young woman was able to reproduce 21 of the 24 cocaine effects, such as talkative, anxious, warm, sociable, altruistic, self-confident, and so on. In the second study [27], five cocaine addicts subjected to a rehabilitation treatment period were able to reproduce many cocaine effects during a single session, including physical sensations like lightness of the body, fluttering in the stomach, sweaty hands, trembling hands, and so on, as well as increased general activation and euphoria and reduced sedation and depression.

All this indicates that SRT can be a very useful technique for producing genuine cocaine effects.

### 3.2. Participants, Materials, and Methods

All the people in the different groups (G1, G2, G3, and G4) of study 1 participated in a relaxation session, where SRT was applied. A subgroup of them (cocaine users) also participated in a second session to reproduce the effects of this drug. The instrument use was the 10 HTSS, which was validated in study 1.

Sessions were carried out in small groups with no more than five participants. At the beginning of the sessions, the participants filled in the 10 HTSS in the state format. They did the same at the end of the session immediately after applying SRT by providing them with relaxing suggestions. This is a pre-post nonequivalent group experimental design.

The next week, 33 cocaine users from G3 and G4 participated in a second session, where SRT was applied to reproduce cocaine-related sensations. As these people had participated in both sessions, this was a within-group experimental design that allowed to compare two different conditions (relaxation and cocaine effects) using the same technique (SRT) on happiness scores.

In this session, cocaine users were trained to apply specific strategies to reproduce the effects of cocaine, such as (1) closing your eyes and remember one of your most recent experiences of drug use, who you were with, the atmosphere, the music, etc.; (2) using a "non-deceiving placebo", that is, put some white powder (sugar or bicarbonate of soda) on the table as a visual stimulus; (3) stage the drug-use ritual by pretending you are snorting the white powder; (4) share this session with the friends you usually use drugs with; (5) create an atmosphere, like by changing the light or putting on some music you associate with the drug; (6) close your eyes and repeat the word "cocaine" in a whisper; (7) describe the drug use sequence (snort then wait for the effects to start, the first effects appear, I can notice them more clearly, etc.); and (8) read the list of effects that you wrote down and repeat the sensations in a whisper.

For the cocaine group, the main inclusion criterion was that the participants had used cocaine at least once in their lives. Additionally, an exclusion criterion was that they did not compulsively take cocaine in the last 12 months, specially no more than six times in the last month, because it is not a study to treat cocaine addiction, but to use the drug to improve the psychological skills in a group of volunteers from the general population. Thus, in this study, no diagnostic criteria about addiction were used.

The participants signed an informed consent form, in which they were encouraged to answer honestly about their drug use.

In Table 4, the levels of drug use in the participants in the last 12 months and in the last month for all the 33 cocaine user participants are shown.

**Table 4.** Levels of drug use in participants in the last 12 months and in the last month (as frequencies) (*n* = 33).

| | How Often in Your Life | | | | How Often in the Last 12 Months | | | | How Often in the Last Month | | | |
|---|---|---|---|---|---|---|---|---|---|---|---|---|
| | 0 | 1–5 | 6–30 | >30 | 0 | 1–5 | 6–30 | >30 | 0 | 1–5 | 6–30 | >30 |
| Cannabis | 0 | 1 | 1 | 31 | 4 | 2 | 8 | 22 | 6 | 7 | 1 | 19 |
| Ecstasy | 3 | 10 | 9 | 11 | 12 | 12 | 4 | 2 | 25 | 5 | 0 | 0 |
| Cocaine | 0 | 8 | 9 | 16 | 7 | 17 | 8 | 1 | 21 | 12 | 0 | 0 |
| Amphetamine | 6 | 12 | 3 | 12 | 7 | 7 | 10 | 3 | 14 | 10 | 3 | 0 |

We can see that only one participant used cocaine more than 30 times in the last year, and none used this drug more than 5 times in the last month. This may mean that most could have trouble remembering and, therefore, reproducing its effects.

*3.3. Results*

First, regarding the relaxing session for all the participants, a one-way ANOVA was performed to determine if there were any differences in the mean scores for all the variables among the four groups.

Below, Table 5 shows the results of the one-way ANOVA, the means and standard deviations (SDs) for all the groups, the overall F-value of the ANOVA, and the corresponding *p*-value, eta-squared partial (effect size), and observed power (1 − β).

**Table 5.** One-factor ANOVA for all the groups (G1, G2, G3, G4) for the study variables.

| | Group | *Mean* | *SD* | *F* | *p* | $\eta^2$ | 1 − β |
|---|---|---|---|---|---|---|---|
| OHQ | G1 | 35.3 | 6.2 | | | | |
| | G2 | 35.6 | 5.9 | 0.04 | 0.986 | 0.001 | 0.05 |
| | G3 | 35.6 | 5.5 | | | | |
| | G4 | 35.9 | 5 | | | | |
| Happiness-Trait | G1 | 36.59 | 8.90 | | | | |
| | G2 | 37.87 | 6.58 | 0.94 | 0.420 | 0.022 | 0.25 |
| | G3 | 35.64 | 6.67 | | | | |
| | G4 | 38.30 | 6.10 | | | | |
| Happiness-State | G1 | 40.03 | 6.42 | | | | |
| | G2 | 36.53 | 6.47 | 1.51 | 0.214 | 0.035 | 0.40 |
| | G3 | 38.27 | 7.24 | | | | |
| | G4 | 38.91 | 7.60 | | | | |
| Extraversion | G1 | 8.04 | 2.94 | | | | |
| | G2 | 8.80 | 3.01 | 3.02 | 0.032 | 0.068 | 0.71 |
| | G3 | 9.76 | 2.01 | 3.01 * | 0.036 | | |
| | G4 | 9.74 | 1.83 | | | | |
| Neuroticism | G1 | 4.22 | 3.47 | | | | |
| | G2 | 4.54 | 3.09 | 0.49 | 0.386 | 0.012 | 0.15 |
| | G3 | 4.43 | 3.20 | | | | |
| | G4 | 3.57 | 3.13 | | | | |
| Psychoticism | G1 | 1.67 | 1.54 | | | | |
| | G2 | 2.10 | 1.62 | 12.33 | <0.001 | 0.230 | 1 |
| | G3 | 4.00 | 2.46 | 11.50 * | <0.001 | | |
| | G4 | 4.43 | 2.77 | | | | |

* A Welch test was conducted because groups had unequal variances for the extraversion and psychoticism variables. Thus, these data violate the homogeneity of variance assumption. Note: G1: never used illegal drugs (*n* = 27); G2: only consumed cannabis as an illegal drug (*n* = 41); G3: moderate drug users (*n* = 37); G4: regular and heavy drug users, especially stimulant drugs (*n* = 23).

The ANOVA test found a significant difference in groups for extraversion ($F(3, 127) = 3.01$, $p = 0.036$, $\eta^2 = 0.068$, $1 - \beta = 0.71$) and psychoticism ($F(3, 127) = 11.50$, $p < 0.001$, $\eta^2 = 0.230$, $1 - \beta = 1$). This corresponds to a medium and high effect level for extraversion and psychoticism, respectively. Pairwise multiple comparisons were performed with Bonferroni fitting for both variables using the Games–Howell post hoc test because equality of variances was not assumed. No differences were found among all the groups for the extraversion variable. For the psychoticism variable, G3 and G4 (moderate, regular + heavy drug users) scored significantly higher than G1 (non-illegal drug users) ($p < 0.001$) 95%CI [−3.66, −1.01] and ($p < 0.001$) 95%CI [−4.53, −1.01], respectively, and G2 (only consume cannabis as an illegal drug) ($p < 0.001$) 95%CI [−3.16, −0.64] and ($p = 0.004$) 95%CI [−4.05, −0.62], respectively.

A mixed analysis of variance, also named a split-plot ANOVA, was used to test for differences in all the groups (between subjects factor) while obtaining repeated measures of participants (within subjects factor). Particularly, this repeated measures ANOVA was performed to compare the effect of relaxation instructions by applying SRT to increase or reduce happiness among the four groups.

A summary of the repeated measures ANOVA results for the happiness state for all participants in this study ($N = 128$) under the RELAXATION condition are presented in Table 6.

**Table 6.** Repeated measures ANOVA results for the happiness state for all the participants in this study ($N = 128$) under the RELAXATION condition.

| | Mean (SD) | | | *F* | *p* | $\eta^2$ | $1 - \beta$ |
|---|---|---|---|---|---|---|---|
| | **Before** | **After** | | | | | |
| G1 | 40.03 (6.42) | 40.48 (6.12) | Happiness | 8.49 | 0.004 | 0.06 | 0.82 |
| G2 | 36.53 (6.47) | 38.56 (6.11) | Happiness × Grup | 0.56 | 0.635 | 0.01 | 0.16 |
| G3 | 38.27 (7.24) | 40.13 (5.75) | Grup | 1.31 | 0.271 | 0.03 | 0.34 |
| G4 | 38.91 (7.60) | 40.26 (5.23) | | | | | |

Note: G1: never used illegal drugs ($n = 27$); G2: only consumed cannabis as an illegal drug ($n = 41$); G3: moderate drug users ($n = 37$); G4: regular and heavy drug users, especially stimulant drugs ($n = 23$).

Mauchly's W test was not calculated when a factor had two levels. In this case, the multivariate analysis results are shown.

Box's test of equality of covariance matrices was 10.500 ($p = 0.337$), and Levene's test of equality of variances among groups for scores BEFORE and AFTER were 0.838 ($p = 0.476$) and 0.417 ($p = 0.471$), respectively. All these results indicate that this ANOVA met assumptions about variances.

There were no statistically significant differences in happiness among all four groups, but an intra-group statistically significant difference appeared ($F(1, 124) = 8.49$, $p = 0.004$, $\eta^2 = 0.06$, $1 - \beta = 0.82$). By considering that the postscore was generally lower than the prescore, we concluded that SRT slightly increased happiness when taking into account such a small effect size ($\eta^2 = 0.06$).

There were no statistically significant differences in the interaction factor.

Finally, two-way repeated measures ANOVAs from the General Linear Model statistical procedure were performed. The two factors were time (BEFORE and AFTER applying SRT) under two conditions (RELAXATION and COCAINE), with two levels each. This analysis was performed to determine if there were differences in the means scores for the happiness variable between both conditions in the cocaine users group ($n = 33$).

Table 7 shows the results of this ANOVA analysis. The primary purpose of a two-way repeated ANOVA is to understand if there is an interaction between two factors of the dependent variable, which is way we present the results of interaction.

**Table 7.** Two-way repeated measures ANOVA of the happiness variable.

| | | *Mean* (SD) | | | *F* | *p* | $\eta^2$ | $1 - \beta$ |
|---|---|---|---|---|---|---|---|---|
| Happiness | Relaxation | 36.8 (7.9) | 38.5 (5.3) | FACTOR 1 | 3.99 | 0.054 | 0.11 | 0.49 |
| | Cocaine | 36.3 (5.9) | 42.1 (5.5) | FACTOR 2 | 24.55 | <0.001 | 0.43 | 0.99 |
| | | | | Condition × Time | 7.99 | 0.008 | 0.20 | 0.78 |

There was a significant principal effect for Factor 2 ($F(1, 32) = 24.55$, $p < 0.001$, $\eta^2 = 0.43$, $1 - \beta = 0.99$). A statistically significant interaction effect between both conditions was found ($F(1, 32) = 7.99$, $p = 0.008$, $\eta^2 = 0.20$, $1 - \beta = 0.78$), but this effect was small because 0.20 is a small effect size for Cohen's value and power did not reach a value of 8.

Pairwise multiple comparisons with Bonferroni fitting for the mean differences and the *t* test for this group ($N = 33$) between the pre- and post-conditions of each session under the two experimental conditions (relaxation and cocaine) were performed.

Below, in Table 8, the *t* test is included so that the effect of SRT under both conditions is clearer.

**Table 8.** Pairwise multiple comparisons for the mean differences and the *t* test for the cocaine users group ($N = 33$) between the pre- and post-conditions.

| | | *MD* | *ST* | *t* | *p* (-Tailed) | *d* (CI 95%) | $1 - \beta$ |
|---|---|---|---|---|---|---|---|
| Happiness | Relaxation | −1.6 | 1.1 | −1.45 | 0.078 | −0.25 (−0.597, 0.96) | 0.29 |
| | Cocaine | −5.7 | 0.9 | −6.01 | <0.001 | −1.047 (−1.468, −616) | 1 |

SRT significantly increases happiness only for the cocaine condition, where the prescores ($M = 36.3$, $SD = 5.9$) were lower than the post scores ($M = 42.1$, $SD = 5.5$) $t(32) = −6.01$, $p < 0.001$, 95%CI [−1.468, −0.616], $1 - \beta = 1$.

## 4. Discussion

By applying Self-Regulation Therapy (SRT) [22], which is a psychological technique based on classic conditioning, expectancy management, and suggestion, a group of people ($N = 128$) were able to relax during an experimental session, which slightly increased their happiness, as measured in the state format from the previously obtained 10-item Happiness Trait–State Scale (10 HTSS). Happiness increased for everyone regardless of their relation to drug use, or whether they belonged to the non-drug use group or the remaining cannabis only, moderate use, or regular heavy drug use groups.

The most remarkable finding was that for some of them, in a non-clinical sample of cocaine users ($N = 33$) cocaine effects were reproduced during an additional session that, in turn, brought about a marked increase in happiness, especially compared to the increase in happiness achieved during the relaxation session.

In fact, the three hypotheses raised at the beginning have been confirmed, as we will verify and detail below.

SRT has already been used to increase the happiness state, as measured by a euphoria scale [92] and validated as a measure of the happiness state in [21,90] and the happiness scale from the Scale for Mood Assessment (EVEA) [93] by reproducing the effects of stimulant drugs like cocaine, speed, ecstasy, and methylphenidate [90,91,94]. In this study, we used a measurement instrument built for this (the 10 HTSS).

The 10 HTSS measures happiness in the trait and state formats and is composed of two scales with five items each: pleasant activation and depression. This scale is positively and closely related to the Oxford Happiness Questionnaire short-format version and extraversion and negatively to neuroticism (EPQ). No significant differences were found in the groups based on whether or not they were drug users, only cannabis users, or moderate or heavy/regular drug users. The psychoticism dimension was closely related to heavy drug use, especially stimulant drugs.

We have previously referred to controversy about the nature of happiness and whether we can consider it a trait or a transitory state. This controversy about the unstable nature of happiness can be resolved by assuming the Pelechano model [95,96], which considers at least three levels of consolidation of personality characteristics: (1) the basic or trait level; (2) the intermediate or contextual level; and (3) the reactive or state level. From this perspective, it is thought that as happiness lies at an intermediate level of consolidation, it can be considered a relatively stable dimension, as well as a process based on situational changes. This was the approach that we herein followed.

However, why did we select happiness as the outcome variable? As deduced from the first part of this study, happiness was chosen as the objective to be improved because it is a concept that largely summarizes and brings together positive emotions, the increase in which is a well-established objective of any psychotherapy or psychiatric treatment. Moreover, happiness is a popular concept that is well understood by the general population as a desirable goal in life, which contributes to making scientific research results more accessible to the general public.

In this study, we showed that it was possible to increase the state happiness during a single session by reproducing the effects of drugs. Therefore, we wondered whether this increase could continue over time and if it could be used in psychotherapy.

In a single case study [23], SRT was applied to reproduce stimulant ephedrine effects in a 35-year-old man with mild depression. After 2 weeks of treatment, this patient's daily scores had significantly increased on the euphoric scale and lowered on that of depression. These changes were maintained for at least 1 month on follow-up.

In another single case study [24], a patient with moderate depression received treatment based on the reproduction of stimulant methylphenidate effects by using SRT and two doses of the drug (once a week). After 2 weeks, the patient displayed reduced negative emotionality and significantly increased positive emotionality, coping strategies, and happiness. The latter was measured by the Oxford Happiness Questionnaire.

Short-term therapy was applied to 15 people who belonged to a non-clinical sample of users of drugs like amphetamines, cannabis, and ecstasy. This therapy consisted of an intense 2 week training based on the reproduction of positive drug effects using SRT to reduce negative emotionality and to increase coping strategies and positive emotionality. They met these therapeutic objectives and were able to maintain them for 1 month on follow-up [25].

All these results indicate that changes in mood and emotions, and even in happiness, achieved during one SRT session can be increased and maintained for a long time with adequate training and monitoring. Therefore, SRT could be useful as a therapeutic procedure to treat emotional disorders in psychotherapy and psychiatry or to improve emotions and mood in the general population and, ultimately, to provide more stable lasting happiness.

Taking into account that cocaine and methylphenidate produce similar subjective and pharmacological effects [97,98]; that methylphenidate abuse by humans is much less frequent than that of cocaine; that it is particularly executed mainly via the intravenous administration route [99]; and that it is currently prescribed for many disorders like attention deficit/hyperactivity disorder, narcolepsy, traumatic brain injuries, and stroke as an adjunctive agent to treat depression, pain, and so on [100], methyphenidate can be a good candidate for replacing cocaine to apply SRT to improve positive emotionality.

Both cocaine and methylphenidate are in schedule II and are, thus, considered medically useful, but with a potential risk of abuse and dependence. Both could be exchanged in different treatments because methylphenidate is becoming available to overcome legal constraints. For example, even with some inconsistent research findings, the potential of methylphenidate as replacement therapy for cocaine dependence must be considered [101]. In addition, ref. [102] found that oral sustained-release methylphenidate is safe when combined with repeated cocaine doses, and it decreases some positive and reinforcing cocaine effects in cocaine abusers with ADHD. In a small sample of cocaine abusers, SRT

was used to reproduce cocaine effects was more effective in reducing cravings for cocaine than with methylphenidate [27].

This study has certain limitations. First, the 10 HTSS must be validated in bigger samples by relating it to other emotional, mood, and personality scales. Although some CFA indices like CFI, TLI, and RMSEA are suitable, $\chi^2$ does not show a good enough fit to the data. However, as SRT training has produced increased positive emotions, and even an increase in happiness using similar scales [90,91,94], we believe that we can accept the results obtained in this study as being adequate, at least provisionally. Second, as previously pointed out, although there is evidence that the emotional changes produced during one SRT session can increase and remain over time and can, therefore, be used in psychotherapy (i.e., [23–25]), it is necessary to carry out a larger and longer experimental study to verify that SRT application to specifically produce cocaine effects can lead to stable lasting happiness for both the general population and a clinical sample.

In addition to these limitations, the frequent criticism of this type of study is about ethical considerations. Taking cocaine produces adverse effects, in this and other studies, no-one has reported any negative effects during SRT training sessions and in the months after its completion. Another criticism is that reproducing a positive cocaine effect can induce people to take and seek cocaine. On the contrary, clinical evidence and research results indicate that this procedure can reduce cravings and the urge to use drugs, as proven with cocaine and heroin addicts [26,27]. The reproduction of the effects of cocaine was completely safe in all cases. In no case did the therapist offer cocaine to the participants or encourage its use. In fact, they were asked to stop using cocaine, at least for the duration of the study, in order to feel cocaine effects "mentally" with SRT. The participants often stated that if they were able to produce drug effects without taking drugs and by an exclusive psychological technique, they would not need to use them. Therefore, SRT may also represent therapeutic potential for treating addictions. In a recent study [103], SRT was applied to reproduce cocaine effects in eight patients with severe cocaine addiction levels (DSM-5 criteria) in a Spanish institution to treat drug addiction. After 11 sessions, the level of craving of all the participants had significantly lowered, as had their stress caused by COVID-19, but their positive emotionality had increased. In other studies [104,105], drug users who participated in ROC (regulation of craving) task strategy training reported significantly lower cravings when focusing on the negative consequences associated with cocaine and methamphetamine use compared to focusing on positive ones. However, there is an important difference when applying SRT. In this case, drug users are trained to focus on the positive effects of drugs in order to reproduce them on purpose.

On the other hand, in the cited study [103], the cocaine users combined the reproduction of the positive effects of cocaine using SRT in the short term with a focus on the negative effects of cocaine use in the long term, based on reading a list of negative effects of cocaine that they previously wrote. It was explained to them that they should not take cocaine in the future because of its negative effects and also because they have learned a therapeutic strategy to deal with the cravings, which is to replace the real cocaine with its "mental" experience, as if it were a "brain deception".

Another criticism purports to assert that this procedure is based on a reductionist conception that conceives happiness only from a hedonistic approach, as an accumulation of pleasant moments, but it must be recognized that this is a widely accepted interpretation of the Western mentality [40]. Obviously, happiness is a broader concept than that, but we can agree that one essential goal of many psychological treatments is to improve mood and positive emotions to achieve stabler and long-lasting feelings of well-being or happiness. This procedure can, thus, effectively contribute to fulfilling this goal. In line with this, Lyubomirsky et al. [106] claim that the happiness–success (i.e., marriage, friendship, and so on) link exists not only because success makes people happy, but also because a positive affect engenders success. A repeated experience of positive effects can lead to success in life.

The application of SRT for the purposes of a study like this one also has other limitations. First of all, SRT works with people who have at least a good level of suggestibility. Second, it is a procedure based on classic conditioning, which means that the effects diminish over time and the participants are required to attend further sessions to reinforce the effects and to, thus, promote their long-term efficacy. Thirdly, SRT, as applied in this study, requires the participants to be or to have been cocaine users. Otherwise, the participants would have to take cocaine to experience its effects for the first time, which would increase the ethical drawbacks of a study like this.

On the other hand, we have to answer whether SRT is effective in reproducing the effects of cocaine rather than producing only a positive emotion, but not necessarily the effects of cocaine. The aforementioned studies [27,89–91] have found that the participants have accurately reproduced many cocaine effects, including internal heat, restlessness, urge to smoke, clear vision, very talkative, bitter mouth, shaky voice, well-being, etc. In this study, the participants reproduced very similar sensations, but in this case our interest focused on the "imitation" of the emotions that cocaine causes, particularly happiness, and to show only these data.

We can see that only one participant in the group of cocaine users used cocaine more than 30 times in the last year, and none used the drug more than 5 times in the last month. This may mean that most of the participants could have trouble remembering and this could, therefore, reproduce its effects. However, that was not the case.

That means that the effects of cocaine can be faithfully reproduced with SRT regardless of cocaine use frequency and the date of last use. Thus, SRT may facilitate a recall of cocaine effects even after a long period of non-use.

Moreover, during the study, no blood or urine samples were taken to detect the presence of cocaine. Although the participants agreed to sign the informed consent not to take cocaine during the study, we are not absolutely certain that the participants complied with this in all cases.

Finally, it can be simply stated that cocaine is an illegal drug. As we point out above, cocaine can be replaced with other legal drugs, such as methylphenidate, but also with amphetamines, ephedrine, or even caffeine. SRT has been successfully applied to reproduce the stimulating and euphoric effects of all these drugs [23,91,107]. SRT training has even proven effective in reproducing not only the subjective effects of methylphenidate, but also in changing personality and gene expression (c-fos and DRD3 genes) in the same way as methylphenidate [108,109].

## 5. Conclusions

SRT is a powerful technique to take advantage of the benefits that all kinds of drugs, whether they are legal or not, can offer us. Indeed, some scientists claim the medical use of cocaine in psychiatry. For example, Hodge et al. [110] (p. 56) state: "Using cocaine in a controlled, medicinal setting by licensed clinical practitioners could yield promising results from clinical trials in the treatment of a wide variety of psychiatric disorders".

We have previously discussed some limitations of using SRT to reproduce the effects of drugs such as cocaine. Among these limitations, we mention that SRT works with people who have at least a good level of suggestibility and requires that the participants are or have been cocaine users, which can make its psychotherapeutic application difficult and increase the ethical drawbacks of its use.

However, on the other hand, as we have also pointed out previously, the reproduction of the effects of cocaine was completely safe in all cases and can even reduce craving and the urge to use drugs [27].

Cocaine produces intense effects of euphoria, well-being, and happiness, which psychotherapists and psychiatrists should take advantage of. SRT offers us that possibility and we will need to keep exploring in depth and improving scientific research designs to obtain this effect.

The ethical question requires a final reflection in the light of everything exposed so far.

Of course, we have to analyze the best conditions for this procedure to be applied safely and without side effects in greater detail, taking into account the necessary training of therapists and selecting participants or patients who can best and most safely benefit from a procedure like this. The correct use of this procedure in the proper context is an essential condition.

Finally, it should be noted that this procedure opens a new path in psychiatry using drugs or medications (e.g., methylphenidate for treatment-resistant depression, ADHD, chronic fatigue syndrome, etc.) in an alternative and unconventional way, which well deserves more attention and study.

**Funding:** This research received no external funding.

**Institutional Review Board Statement:** Both first and second studies of this article were conducted in accordance with the Declaration of Helsinki, and the protocol was approved by the Ethics Committee of the University of Valencia (Spain) on 15 September 2017 (project identification code: H1499339130100).

**Informed Consent Statement:** Informed consent was obtained from all subjects involved in the study.

**Data Availability Statement:** The data presented in this study are available on request from the corresponding author. The data are not publicly available due to ethical consideration and to preserve the privacy of the participants.

**Acknowledgments:** The author thanks all of the participants in this study.

**Conflicts of Interest:** The author declares no conflict of interest.

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
