# Peer review of "Cocaine Effects without Cocaine: Increasing Happiness with Self-Regulation Therapy in a Single Session"

_2673-5318, doi:10.3390/psychiatryint4030026_

Round 1

Reviewer 1 Report

Thank you for reading your manuscript. Interesting study, a couple of comments see below/ Kindest regards Nina Gårevik

The background is solid and relevant. However, a brief overview of previous research in the field of therapy and treatment of substance use disorders is missing, for examples mindfulness, yoga, cognitive therapy and  reinforcement of,  treatment of addiction with substitutiontherapy  (for example subutex-mindfulness)

The rationale for the study needs to be clarified.
Material och Method:  Suggests a table of when last dose of narcotic preparations was taken before inclusion and intervention? Alternatively, clarify how long before the intervention the estimate of drug use was made, (see table 4)
H
ow were the answers in the substance use scale verified? Were any drug tests done?
Who performed the SRT session, was everyone trained in it and who were the ones who conducted it?

Reviewer 2 Report

Introduction

Overall, the introduction provides a clear overview of the topic and research objectives. The use of cocaine in medicine is outlined, as well as its potential for abuse and negative side effects. The article then proposes the use of Self-Regulation Therapy (SRT) as an alternative to using cocaine to increase positive emotions in a safe and controlled environment.

However, there are a few areas where the introduction could be improved. For example, the article could provide more detail on the SRT and how it works. Additionally, it may be beneficial to provide more context on the importance of measuring happiness and the potential benefits of increasing positive emotions. Overall, the introduction provides a good foundation for the study and raises interesting questions for further research.

First study

The introduction section provides a comprehensive overview of previous research on measuring happiness, including various scales and instruments that have been used.

However, the section could benefit from more explicit connections to the main research question of the study, specifically how the 10 HTSS will be used to test the effect of the SRT on happiness compared to the effects of cocaine. Additionally, some of the references cited in this section could be more clearly tied to the main research objectives.

Methods

Overall, the methods section provides a clear and detailed overview of the study design and methods that will be used to analyze the data.

Results

It is recommended to provide more details on the sample size and characteristics, as well as the rationale for selecting the specific scales used for convergent and discriminant validity. Additionally, it should be considered discussing the implications of the findings and potential future research directions.

Second study

I recommended to provide more background information on the potential risks and ethical considerations associated with reproducing drug effects in a therapeutic setting. Also, the author could consider discussing the limitations of the SRT technique and potential confounding factors that may affect the results.

Participants, Materials, and Methods

This section of the scientific study outlines the participants and procedures utilized in the research. To enhance the section, it is suggested that the author provides more information on the recruitment process and the criteria used for participant inclusion and exclusion. Furthermore, the author could discuss the potential limitations of the study, such as the small sample size and the possibility of recall bias among cocaine users.

Results

Overall, this section appears to be well structured and provides specific details about the statistical analyses carried out on the study data. However, some minor improvements could be made to enhance clarity and readability, for example by proofreading for typographical errors, fixing some grammatical issues, and explaining some technical terms or abbreviations not defined in the section. Overall, this section seems to be of high quality and provides a good basis for further interpretation and discussion of the findings.

Discussion

The discussion section is well-organized and provides an in-depth analysis of the study results and their implications. The author(s) offer a comprehensive overview of the SRT technique used and cite relevant literature to support their arguments. Additionally, potential criticisms of the study, such as ethical concerns, are addressed with reasoned responses. The discussion also explores the potential therapeutic implications of the SRT technique for individuals with emotional disorders and addiction. While some minor grammatical issues and phrasing could be improved for clarity, overall, the discussion section meets the requirements of a well-structured and comprehensive analysis of the research findings, limitations of the study, and future implications.

Conclusions

The conclusion section presents a thought-provoking argument regarding the potential benefits of using the SRT technique to safely replicate the positive effects of drugs, including illegal ones like cocaine. While citing Hodge et al.'s suggestion of exploring the medical use of cocaine in psychiatry, the ethical implications of refusing to utilize SRT to replicate positive drug effects are raised. A more in-depth discussion to support this argument would have been helpful, including references to relevant studies or ethical frameworks. However, the conclusion appropriately reflects the implications of the study and presents intriguing ethical questions for further exploration.

Reviewer 3 Report

A. General Comments:

  I would like to suggest to the authors that they should consider dividing their study into two papers. The current title explains the aim and objective of Study 2. it would be better if the authors remove the first study from this manuscript and publish it separately. It would greatly benefit the readers if the authors only provide information about the psychometric properties of the 10-item Happiness Trait-State Scale (10 HTSS), including reliability and validity, in this paper.

B.Specific Comments:

 Common Introduction:

  1. Page 1,Line 23-24:The statement "but of acceptable medical use, especially as an effective safe anesthetic in surgical nasal, ear and throat procedures" needs to be rephrased as it implies that cocaine could be an alternative safe anesthetic agent. Currently, it is not in practice anywhere in the world.
  2. Page 1, Line 25-26: Add the name of the country in which this law is effective, as not everyone may be familiar with the United States federal law.
  3. Page 2, Line 62-63: The statement "Patients using the SRT can feel euphoria and well-being as often as they wish and it could, thus, increase their happiness" requires a reference to support this claim.
  4. Page 2, Line 92-74: The sentence "The main hypothesis of this study is if it is possible to first obtain a reliable measure of the trait and state of happiness and, second, to increase this emotion by reproducing cocaine effects during a stimulating session rather than a relaxing one" needs improvement for clarity and coherence.

Page 2-8, Lines 75-292: The authors should consider a separate publication for the first study, which focuses on the 10-item Happiness Trait-State Scale (10 HTSS). In the second study, A can provide the psychometric properties of the 10 HTSS.

 Second study: reproducing cocaine effects to increase happiness

1.       Page 8-9, lines 295-324, Introduction: The second study, which aims to reproduce the effects of cocaine to increase happiness, requires improvement in several areas, particularly in the introduction should be rewritten to provide a clear and synchronized explanation of the study's objectives, along with research questions and hypotheses.

2.       Page 9, line 325-354, Participants, Materials and Methods:

i.                    Positive cocaine effects vs. Happiness:  whether the participants experienced the positive effects of cocaine or happiness. It is important to clarify whether the observed effects were an outcome expectancy of cocaine use or an overall sense of happiness.

ii.                  Operational definition for groups G3 and G4: Provide an operational definition of the groups labeled as G3 (moderate drug users) and G4 (regular and heavy drug users). It would be helpful to provide clear criteria or characteristics that distinguish these groups.

iii.                Provide about the participants' diagnosis of cocaine use disorder and the timing of their last drug dose. It is crucial to include information about the diagnostic criteria used and the time frame within which participants had used cocaine.

iv.                 Provide more information about the participants, recruitment process, and the criteria used to determine the group assignments. Additionally, it is important to mention if drug assessments in body fluids were conducted to confirm the presence of cocaine.

v.                   Provide more information about the procedure used for Self-Regulation Therapy and how it was applied to participants to increase happiness in the cocaine group. Specifically, more details are needed on how the therapy sessions were conducted?

vi.                 10-item Happiness Trait-State Scale (10 HTSS) was validated in Study 1. It would be beneficial to provide information about the psychometric properties of the scale, such as reliability and validity, in the paper.

Discussion

i.                    Page 13, line 426: "The most remarkable finding was that for some of them, a non-clinical sample of cocaine users (N=33)." Please explain what is meant by a "non-clinical sample."

ii.                  Page 13, line 47: "cocaine effects during an additional session that brought about a marked increase in happiness." Why did the author select happiness as the outcome variable? Did they assess other cocaine effects as well? Please provide details of these effects. How long did the happiness effect persist?

The conclusions section needs to be rewritten with regard to the concerns raised.

A. General Comments:

  I would like to suggest to the authors that they should consider dividing their study into two papers. The current title explains the aim and objective of Study 2. it would be better if the authors remove the first study from this manuscript and publish it separately. It would greatly benefit the readers if the authors only provide information about the psychometric properties of 10-item Happiness Trait-State Scale (10 HTSS), including reliability and validity, in this paper.

 B.Specific Comments:

 Common Introduction:

  1. Page 1,Line 23-24:The statement "but of acceptable medical use, especially as an effective safe anesthetic in surgical nasal, ear and throat procedures" needs to be rephrased as it implies that cocaine could be an alternative safe anesthetic agent. Currently, it is not in practice anywhere in the world.
  2. Page 1, Line 25-26: Add the name of the country in which this law is effective, as not everyone may be familiar with the United States federal law.
  3. Page 2, Line 62-63: The statement "Patients using the SRT can feel euphoria and well-being as often as they wish and it could, thus, increase their happiness" requires a reference to support this claim.
  4. Page 2, Line 92-74: The sentence "The main hypothesis of this study is if it is possible to first obtain a reliable measure of the trait and state of happiness and, second, to increase this emotion by reproducing cocaine effects during a stimulating session rather than a relaxing one" needs improvement for clarity and coherence.

Page 2-8, Lines 75-292: The authors should consider a separate publication for the first study, which focuses on the 10-item Happiness Trait-State Scale (10 HTSS). In the second study, A can provide the psychometric properties of the 10 HTSS.

Second study: reproducing cocaine effects to increase happiness

1.       Page 8-9, lines 295-324, Introduction: The second study, which aims to reproduce the effects of cocaine to increase happiness, requires improvement in several areas, particularly in the introduction should be rewritten to provide a clear and synchronized explanation of the study's objectives, along with research questions and hypotheses.

2.       Page 9, line 325-354, Participants, Materials and Methods:

i.                    Positive cocaine effects vs. Happiness:  whether the participants experienced the positive effects of cocaine or happiness. It is important to clarify whether the observed effects were an outcome expectancy of cocaine use or an overall sense of happiness.

ii.                  Operational definition for groups G3 and G4: Provide an operational definition of the groups labeled as G3 (moderate drug users) and G4 (regular and heavy drug users). It would be helpful to provide clear criteria or characteristics that distinguish these groups.

iii.                Provide about the participants' diagnosis of cocaine use disorder and the timing of their last drug dose. It is crucial to include information about the diagnostic criteria used and the time frame within which participants had used cocaine.

iv.                 Provide more information about the participants, recruitment process, and the criteria used to determine the group assignments. Additionally, it is important to mention if drug assessments in body fluids were conducted to confirm the presence of cocaine.

v.                   Provide more information about the procedure used for Self-Regulation Therapy and how it was applied to participants to increase happiness in the cocaine group. Specifically, more details are needed on how the therapy sessions were conducted?

vi.                 10-item Happiness Trait-State Scale (10 HTSS) was validated in Study 1. It would be beneficial to provide information about the psychometric properties of the scale, such as reliability and validity, in the paper.

Discussion

i.                    Page 13, line 426: "The most remarkable finding was that for some of them, a non-clinical sample of cocaine users (N=33)." Please explain what is meant by a "non-clinical sample."

ii.                  Page 13, line 47: "cocaine effects during an additional session that brought about a marked increase in happiness." Why did the author select happiness as the outcome variable? Did they assess other cocaine effects as well? Please provide details of these effects. How long did the happiness effect persist?

The conclusions section needs to be rewritten with regard to the concerns raised.

Round 2

Reviewer 3 Report

The authors have addressed valid suggestions and improved their manuscript, and now it may be considered in its current form.

Author Response

Thank you very much for your review. If you would like us to get in touch, I will be happy to interchange ideas and research.